# BOLD Coupling between Lesioned and Healthy Brain Is Associated with Glioma Patients’ Recovery

**DOI:** 10.3390/cancers13195008

**Published:** 2021-10-06

**Authors:** Rafael Romero-Garcia, Michael G. Hart, Richard A. I. Bethlehem, Ayan Mandal, Moataz Assem, Benedicto Crespo-Facorro, Juan Manuel Gorriz, Gladstone Austin Amos Burke, Stephen J. Price, Thomas Santarius, Yaara Erez, John Suckling

**Affiliations:** 1Department of Psychiatry, University of Cambridge, Cambridge CB2 0SZ, UK; mgh40@cam.ac.uk (M.G.H.); rb643@medschl.cam.ac.uk (R.A.I.B.); asm253@georgetown.edu (A.M.); gorriz@ugr.es (J.M.G.); js369@cam.ac.uk (J.S.); 2Department of Medical Physiology and Biophysics, Instituto de Biomedicina de Sevilla (IBiS), HUVR/CSIC/Universidad de Sevilla, 41013 Sevilla, Spain; 3MRC Cognition and Brain Sciences Unit, University of Cambridge, Cambridge CB2 7EF, UK; Moataz.Assem@mrc-cbu.cam.ac.uk (M.A.); yaara.erez@mrc-cbu.cam.ac.uk (Y.E.); 4Department of Psychiatry, Instituto de Investigación Sanitaria de Sevilla, IBiS, Hospital Universitario Virgen del Rocio, CIBERSAM, 41013 Sevilla, Spain; benedicto.crespo@unican.es; 5Department of Signal Theory, Networking and Communications, Universidad de Granada, 18071 Granada, Spain; 6Department of Paediatric Haematology, Oncology and Palliative Care, Addenbrooke’s Hospital, Cambridge CB2 0QQ, UK; ab2359@cam.ac.uk; 7Academic Neurosurgery Division, Department of Clinical Neurosciences, University of Cambridge, Cambridge CB2 0QQ, UK; sjp58@cam.ac.uk (S.J.P.); ts381@cam.ac.uk (T.S.); 8Department of Physiology, Development and Neuroscience, University of Cambridge, Cambridge CB2 0SZ, UK; 9Faculty of Engineering, Bar-Ilan University, Ramat Gan 5290002, Israel; 10Behavioural and Clinical Neuroscience Institute, University of Cambridge, Cambridge CB2 0SZ, UK; 11Cambridge and Peterborough NHS Foundation Trust, Cambridge CB21 5EF, UK

**Keywords:** global signal, brain tumours, functional MRI, neurosurgery, cognitive recovery

## Abstract

**Simple Summary:**

Glioma, a type of brain tumour, affects not only the function of immediately adjacent brain tissue but also that in more distant areas, potentially impacting cognitive function after its surgical removal. Here, 17 patients with glioma had brain scans and tests of cognitive function during treatment and recovery. We investigated the effects of glioma on the brain, and what happens during recovery, using the brain’s “global signal” detected with magnetic resonance imaging (MRI). We found that the signal from gliomas was synchronised with the global signal in all patients and that this synchronisation was associated with the recovery of cognition after surgery. Specifically, patients with a greater reduction in glioma–global signal synchronisation following surgery were more likely to have a larger number of newly acquired cognitive difficulties. Together, these results suggest that the interaction between gliomas and the brain can predict how patients recover their cognitive abilities, which is important for their quality of life.

**Abstract:**

Predicting functional outcomes after surgery and early adjuvant treatment is difficult due to the complex, extended, interlocking brain networks that underpin cognition. The aim of this study was to test glioma functional interactions with the rest of the brain, thereby identifying the risk factors of cognitive recovery or deterioration. Seventeen patients with diffuse non-enhancing glioma (aged 22–56 years) were longitudinally MRI scanned and cognitively assessed before and after surgery and during a 12-month recovery period (55 MRI scans in total after exclusions). We initially found, and then replicated in an independent dataset, that the spatial correlation pattern between regional and global BOLD signals (also known as global signal topography) was associated with tumour occurrence. We then estimated the coupling between the BOLD signal from within the tumour and the signal extracted from different brain tissues. We observed that the normative global signal topography is reorganised in glioma patients during the recovery period. Moreover, we found that the BOLD signal within the tumour and lesioned brain was coupled with the global signal and that this coupling was associated with cognitive recovery. Nevertheless, patients did not show any apparent disruption of functional connectivity within canonical functional networks. Understanding how tumour infiltration and coupling are related to patients’ recovery represents a major step forward in prognostic development.

## 1. Introduction

Surgical resection with adjuvant chemo- and radio-therapy is employed in the management of patients with treatments to delay brain tumours and their progression and improve survival in patients with diffuse glioma. Nevertheless, a large proportion of patients with glioma suffer cognitive impairments, such as memory, attention, language and executive deficits, that can significantly impair their quality of life [1,2]. A wide variety of clinical and demographic factors contribute to individual differences in neurocognitive outcomes of brain tumour patients [3,4], including psychological distress, tumour characteristics, tumour-related epilepsy and therapeutic interventions (surgery, chemoradiotherapy, antiepileptics or corticosteroids) [2]. Despite cognitive functioning now being recognised as an independent prognostic factor [5], little is known about how cognition is affected by tumour–brain functional interactions.

Blood oxygenation level-dependent (BOLD) functional magnetic resonance imaging (fMRI) detects changes in an endogenous paramagnetic contrast agent (deoxyhaemoglobin) that is sensitive to neuronal activation. However, various other anatomical, physiological and imaging parameters contribute to the BOLD signal. For example, its dependency on oxygenation level and cerebral blood volume [6] makes the resulting signal particularly susceptible to vascular fluctuations [7]. Furthermore, the average BOLD signal intensity across cortical grey matter (GM), defined as the global signal (GS), is affected by non-neuronal sources, such as head motion [8] and respiratory and cardiac cycles [9]. Nevertheless, a growing body of literature has shown that the GS carries information about widespread neural activity with biological relevance [10]. Evidence from non-human primate models shows that local field potentials from single electrodes are correlated with resting-state BOLD signal measures across the cortex [11]. Simultaneous recordings of EEG-fMRI in humans have revealed that broadband fluctuations in EEG power are spatially correlated with fMRI, with a 5 s time lag [12]. Using a similar methodology, Wong et al. [13] found that decreases in GS amplitude are associated with increases in vigilance, which is consistent with previously observed associations between the GS and caffeine-related changes [14]. Moreover, the GS recapitulates well-established patterns of large-scale functional networks that have been associated with a wide variety of behavioural phenotypes [15]. However, the relationship between GS alterations and cognitive disruption in neurological conditions remains, at best, only partially understood.

Despite structural MRI being routinely used for brain tumour detection and monitoring, the clinical applications of fMRI to neuro-oncology are currently limited. A growing number of surgical units are exploiting fMRI for presurgical mapping of speech, movement and sensation to reduce the number of post-operative complications in patients with brain tumours and other focal lesions [16,17,18]. Recent fMRI studies have demonstrated the potential of BOLD for tumour identification and characterisation [19]. The abnormal vascularisation, vasomotion and perfusion caused by tumours have been exploited for performing accurate delineation of gliomas from surrounding normal brain [20]. Thus, fMRI, in combination with other advanced MRI sequences, represents a promising approach for a better understanding of intrinsic tumour heterogeneity and its effects on brain function.

Supplementing traditional histopathological tumour classification, BOLD fMRI can provide insights into the impact of a tumour on the rest of the brain (i.e., beyond the tumour’s primary location). Glioblastomas reduce the complexity of functional activity not only within and close to the tumour but also at long ranges [21]. Alterations of functional networks before glioma surgery have been associated with increased cognitive deficits independent of any treatment [22]. One potential mechanism of tumoural tissue influencing neuronal activity and thus cognitive performance is through alterations in oxygenation level and cerebral blood volume [23]. However, it has been suggested that the long-distance influence of tumours in brain functioning is independent of hemodynamic mechanisms [24] and that it is associated with overall survival [25]. To date, no study has explored how BOLD interactions between tumour tissue and the rest of the brain affect the GS, nor how this interaction might impact cognitive functioning.

In this longitudinal study, we prospectively assessed a cohort of patients with diffuse glioma pre- and post-operatively and at 3 and 12 months during the recovery period. Our primary aim was to understand the impact of the tumour and its resection on whole-brain functioning and cognition. The secondary aims of this research were to assess: (i) the GS topography and large-scale network connectivity in brain tumour patients, (ii) the BOLD coupling between the tumour and brain tissue and iii) the role of this coupling in predicting cognitive recovery. Given the widespread effects of tumours on functional brain networks, we hypothesised that these effects would be observable in the GS and, specifically, that the topography of its relationship with regional signals would be altered compared to patterns seen in unaffected control participants. The GS is known to be associated with cognitive function, and, thus, we also hypothesise that changes in the topographic relationship of the GS would be related to changes in cognition experienced by patients as a result of their surgical treatment.

## 2. Materials and Methods

### 2.1. Sample

This single-centre, prospective cohort study was approved by the Cambridge Central Research Ethics Committee (Reference number 16/EE/0151). Patients with a typical appearance of a diffuse glioma were identified at adult neuro-oncology multidisciplinary team (MDT) meetings at Addenbrooke’s Hospital (Cambridge, UK). A consultant neurosurgeon directly involved in the study identified potential patients based on the outcome of the MDT discussion. All patients gave written informed consent. 

The inclusion criteria were the following: (i) participant is willing and able to give informed consent for participation in the study; (ii) imaging is evaluated by the MDT and judged to have typical appearances of a diffuse non-enhancing glioma; (iii) Stealth MRI is obtained (a routine neuronavigation MRI scan performed prior to surgery); (iv) World Health Organisation (WHO) performance status 0 or 1; (v) age between 18 and 80 years; (vi) tumour located in or near eloquent areas of the brain, i.e., regions that according to the MDT may be critical for speech comprehension and articulation, such as the superior temporal lobe and inferior frontal gyrus; and (vii) patient undergoing awake surgical resection of a diffuse glioma. This last inclusion criterion was adopted to collect additional intraoperative electrocorticography data, which have been reported separately [26]. Participants were excluded if any of the following applied: (i) concomitant anti-cancer therapy, (ii) history of previous malignancy (except for adequately treated basal and squamous cell carcinoma or carcinoma in situ of the skin) within 5 years and (iii) previous severe head injury. 

Eighteen patients aged 22–56 years (8 females) were approached to take part. All consented, but one participant subsequently withdrew due to not being able to tolerate the MRI environment (see Appendix A for demographics). Final histological diagnoses revealed different grades of glioma: WHO-I *n* = 2, WHO-II *n* = 7, WHO-III *n* = 5 and WHO-IV *n* = 3. Adjuvant chemoradiotherapy was performed in 12 patients. Each patient was scanned up to four times: before surgery (preop), within 72 h after surgery (postop) and at 3 and 12 months after surgery (month-3 and month-12). 

Data from patients with diffuse glioma collected here were complemented with two publicly available datasets. First, there were 653 cognitively healthy controls (HCs; age range = 18–88 years) from the Cambridge Centre for Aging and Neuroscience (Cam-CAN) [27]. Inclusion/exclusion criteria and MRI processing protocols are described elsewhere [28]. Second, there were structural MRI data and tumour masks of 335 patients with glioma (no fMRI available) from the Multimodal Brain Tumour Image Segmentation Challenge 2019 (BraTS; http://braintumorsegmentation.org, accessed on 30 June 2019). Pre-processing and tumour frequency estimation are described in [29]. The following processing and analyses steps refer exclusively to data from 17 patients with diffuse glioma.

### 2.2. MRI Data Acquisition and Pre-Processing

MRI data from diffuse glioma patients were acquired at the Wolfson Brain Imaging Centre (University of Cambridge) using a Siemens Magnetom Prisma-fit 3 Tesla MRI scanner and 16-channel receive-only head coil (Siemens AG, Erlangen, Germany). A T1-weighted MPRAGE sequence was acquired using the following parameters: repetition time (TR) = 2300 ms, echo time (TE) = 2.98 ms, flip angle (FA) = 9°, 1 mm^3^ resolution, field of view (FOV) = 256 × 240 mm^2^, 192 contiguous slices and acquisition time of 9 min and 14 s. During the same scanning session, we acquired resting-state (eyes closed) fMRI with a BOLD-sensitive sequence: TR = 1060 ms, TE= 30 ms, acceleration factor = 4, FA = 74°, 2 mm^3^ resolution, FOV = 192 × 192 mm^2^ and acquisition time of 9 min and 10 s. fMRI pre-processing was based on independent component analysis (ICA) performed with FSL MELODIC. Noise components were identified and removed using ICA-FIX [30] with training specific to this dataset [31]. Additional processing steps included slice timing correction, bias field correction, rigid body motion correction, normalisation by a single scaling factor and smoothing to 5 mm fixed-width half-maximum. We focused on the physiologically relevant frequency range by using wavelet filtering that retained the BOLD oscillations in the frequency range 0.03–0.12 Hz (wavelet scales 3 and 4) [32].

### 2.3. Lesion Masking, Image Co-Registration, Parcellation and Time-Series Extraction

Masks of the pre-operative tumour and follow-up lesion (reflecting, for example, resected tissue, residual tumour, post-operative oedema or gliosis) were created using a semi-automated procedure. For each participant, initially, an experienced neurosurgeon (MGH) manually delineated the tumour on the pre-operative T1-weighted image slices and the signal change adjacent to the resection cavity on the follow-up images. However, the accuracy of manually defined masks is limited by the human rater’s view. Therefore, we further refined each mask using the Unified Segmentation with Lesion toolbox (https://github.com/CyclotronResearchCentre/USwithLesion, accessed on 31 April 2020), which accounts for lesion distortion by adding a subject-specific probability map before spatially warping from the subject to reference space where tissue probability maps are predefined [33].

The image of the brain then underwent enantiomorphic filling of the lesioned region following a cortical reconstruction using FreeSurfer 6.0. In brief, each image was subjected to skull stripping, segmentation (i.e., identification of tissue compartments) and reconstruction of the pial surface and grey–white matter boundary. The Desikan–Killiany atlas implemented in Freesurfer was subdivided into 318 contiguous cortical parcels of an approximately equal area of 500 mm^2^ using a subparcellation algorithm previously described [34]. The resulting parcellation was transformed from fsaverage standardised space to native space using surface-based non-linear registration. Sixteen subcortical regions were added to the cortical parcels resulting in a brain parcellation with 334 regions. 

Regional tumour frequency was defined as the ratio of patients with a tumour covering at least 50% of each parcel. Inter-regional distances to the tumour boundary, as identified by the tumour mask, were estimated as the geodesic distance of the shortest path constrained by the white matter. fMRI was linearly co-registered (6 degrees of freedom) to the T1 image using ANTs (http://stnava.github.io/ANTs/, accessed on 31 October 2016). The resulting inverse transformation was used to map the T1-based parcellation into the fMRI space for the extraction of the average time series of each parcel. 

Framewise displacement (FD), a measure of head movement during scanning, was computed for each timepoint. To mitigate the potentially confounding effects of head motion, frames with FD > 0.4 were identified as outliers. The frame before and the two frames after the outliers were also considered outliers due to the delayed effect of motion in the BOLD signal. Frames labelled as outliers were removed from the time series. One scan with more than 50% of outliers was completely removed due to a poor signal-to-noise ratio. After exclusions and losses to follow-up, 55 MRIs and 31 neuropsychological assessments acquired from diffuse glioma patients were included in the analyses (see Appendix A).

We deployed the mapping of large-scale canonical functional brain networks defined in Yeo et al. (2011) [35]. This atlas was created by clustering functionally coupled regions in 1000 young, healthy adults. Regions delimited on the 7-Network version were used for calculating the functional correlation within each canonical network. 

### 2.4. Neuropsychological Assessment

Patients were cognitively assessed two weeks before surgery and between two and five weeks after surgery. The neuropsychological assessment comprised 26 independent measures of cognitive function across eight domains: verbal memory (Adult Memory and Information Processing Battery Task—AMIPB—story, immediate and delayed recall; Brain Injury Rehabilitation Trust Memory and Information Processing Battery—BIMPB—word and list recognition), nonverbal memory (BIMPB complex figure and design learning), verbal skills (premorbid functioning, graded naming test, syntactic speech comprehension, letter and semantic fluency), nonverbal skills (BIMPB complex figure, object decision, number location and cube analysis), attention (Wechsler Adult Intelligence Scale—WAIS—IV digit span forward and backward) and executive function (Brixon, initiation, inhibition time and score), using previously validated tests [36,37,38]. Testing took approximately 2–3 h to complete and was administered by a neuropsychologist in a clinical setting.

Following a convention used in previous studies [39,40], a deficit was defined as performance two standard deviations below the mean of a reference population on any particular test or test component [41]. The total number of deficits was defined as the sum of tests where a given patient scored below the threshold. The number of acquired cognitive deficits (Δ Total cognitive deficits) was computed as the difference between the total number of deficits during post-operative follow-up assessment minus the deficits before surgery. Thus, Δ Total cognitive deficits above zero represent patients that acquired new deficits during treatment (cognitive deterioration), while scores below zero correspond with patients who have a reduced number of deficits (cognitive recovery). 

### 2.5. BOLD Signal Extraction from Tissue Compartments and Analysis

BOLD signals were extracted and averaged across voxels for several tissue compartments: (i) tumour/lesion, as defined by the semiautomatic delineation procedure (referred to as ‘tumour ipsi’); (ii) cerebrospinal fluid (CSF); (iii) white matter (WM); (iv) brain tissue contralateral to the tumour (referred to as ‘tumour contra’); and (v) grey matter (GM) within the 318 cortical regions and 16 subcortical structures (thalamus, caudate, putamen, pallidum, hippocampus, amygdala, accumbens and ventral diencephalon) defined by the atlas, excluding the region of the tumour/lesion. The average BOLD signal extracted from this GM constituted the GS. The association, β, was the slope of the line relating to the BOLD signals derived from two different compartments; see Figure 1 for an illustrative flowchart. As patients present tumours in different brain locations, a comparison of β in lesioned tissue across patients was performed after normalising the ipsilateral values by the contralateral values ($\hat{\beta }$).

Traditional parametric methods for relating brain maps ignore the inherent spatial auto-correlation of brain features (i.e., the data independence assumption is violated). To avoid inflated estimations of significance values, correspondence between regional β maps and tumour frequency maps was statistically tested by generating 10,000 random rotations (i.e., spins) of the cortical parcellation to estimate the distribution of β under the null hypothesis. This process provided a reference distribution for significance testing (P*_spin_*) of brain feature associations across regions while controlling for spatial contiguity and hemispheric symmetry of the cortical surface [42]. 

Age, tumour volume and treatment (surgery alone vs. surgery plus adjuvant therapy) were regressed out before performing statistical testing of the association between longitudinal $\hat{\beta }$ changes and cognitive recovery. Some statistical tests were repeatedly performed across patients, tissue types (i.e., GM, WM, CSF, tumour ipsi and tumour contra) and assessments (i.e., pre-operative, post-operative, 3 and 12 months). In those cases, *p*-values were corrected for multiple comparisons using the Benjamini–Hochberg false discovery rate (FDR < 0.05) to reduce the likelihood of false positive findings.

## 3. Results

### 3.1. GS Topography Is Associated with Tumour Occurrence

First, we tested whether tumour occurrence was related to the regional topography of the BOLD signal (GS topography). GS topography was computed as the slope of the line, β, relating the GS (defined as the average BOLD signal across all GM cortical and subcortical voxels, excluding tumour) and each regional BOLD signal. The GS topography in HCs from the Cam-CAN dataset showed the strongest regional associations in the medial occipital cortices, while the insula and prefrontal cortices had the weakest associations (Figure 2, top, left), replicating the spatial pattern previously reported [8,15]. These regional β values were negatively correlated with the corresponding regional glioma occurrence in both the BraTS dataset (R^2^ = 0.14; P*_spin_*= 0.026; Figure 2) and the longitudinal dataset of patients with diffuse glioma (R^2^ = 0.20; P*_spin_*= 0.016; Figure 2); that is, tumours are preferentially located in regions showing low coupling with the GS. 

### 3.2. Brain Tumour Patients Have a Long-Term Alteration of GS Topography

We then explored the spatial reorganisation of the GS topography in patients with brain tumours. In contrast with the regional β maps observed in HCs, patients with diffused gliomas showed a relative increase in GS coupling in the posterior occipital and parietal cortex (Figure 3A). After averaging across participants, variations in β across regions were significantly reduced in brain tumour patients (SD = 0.23, β ∈ [0,1.3]) compared to HCs (SD = 0.43, β ∈ [0.17,2.61], F*_stat_* = 0.30, *p* < 10^−12^). Compared with HCs, regional β maps of the brain tumour patients showed reductions in the medial occipital cortex and posterior cingulate (Figure 3B). Conversely, patients had a relatively increased β in association cortices, such as the prefrontal cortex and the inferior parietal lobe. These differences, both decreases and increases, were preserved after tumour resection (post-operative) and during recovery (3 and 12 months follow-up), suggesting that the tumour induces a long-term reorganisation of BOLD dynamics.

### 3.3. Tumour BOLD Time Series Is Coupled with GS

We next aimed to determine how different brain tissues are coupled with the GS. The GS was significantly correlated with the BOLD signal derived from each of the tissue compartments: CSF, WM, tumour and cortical regions contralateral to the tumour (*p* < 0.05 for all patients and tissues except for four cases; FDR corrected; see Appendix A). However, the GS was differentially associated with the BOLD signal depending on the tissue compartment. The value of β between the GS and tumour BOLD signal was significantly higher than that between (i) the GS and CSF, (ii) the tumour signal and CSF and (iii) the GS and WM (non-parametric Wilcoxon test; all *p* < 0.05; FDR corrected; Figure 4A). BOLD signals from CSF and WM are often considered to contain limited physiological information, and, thus, the relative elevation of β suggests that strong GS–tumour BOLD coupling potentially represents the functional integration of the tumour with the brain. However, the value of β between the GS and tumour signal was significantly lower than that between the GS and BOLD signals derived from the cortical regions contralateral to the tumour (*p* = 0.015; FDR corrected). Regions that were closer to the tumour showed lower associations with the GS (Figure 4B), with regions in the hemisphere contralateral to the tumour also showing a similar distance-to-tumour effect (Figure 4B) reflecting the preferential occurrence of brain tumours in regions with low GS coupling in HCs (as shown in Figure 2A). Thus, although the tumour BOLD signal may include contributions from functional processing, they are not to the same degree as those in unaffected regions.

### 3.4. Lesion–GS Coupling Is Preserved during Recovery and Is Associated with Cognition

We subsequently tested whether tissue lesioned by surgical resection (e.g., cavity, oedema and residual tumour) was also coupled with the GS and its potential associations with cognitive recovery. Pre-operative tumour–GS coupling ($\beta_{preop}^{ipsi}$) was significantly higher than the coupling between the GS and the lesioned tissue that remained after tumour resection ($\beta_{postop}^{ipsi\;}$, non-parametric Wilcoxon test; *p* = 0.0025; FDR corrected) but then did not significantly change during follow-up ($\beta_{follow-up}^{ipsi}$; 3 months *p* = 0.12 and 12 months *p* = 0.08; FDR corrected; Figure 5A). To account for the differential location of the lesioned tissue, β values were normalised before comparing patients. The normalised lesion–GS coupling, $\hat{\beta }$, was defined as the ratio between ipsilateral (Figure 5A, left) and contralateral (Figure 5A, middle) values of β. $\hat{\beta }$ was reduced after surgery (Figure 5A, right), but significance did not survive correction for multiple comparisons across assessments (P_uncorrected_ = 0.02; P_FDR-corrected_ = 0.05). The rate of change of normalised coupling during the recovery period, Δ$\hat{\beta }$, was significantly associated with ${\hat{\beta }}_{preop}$; that is, patients with higher tumour–GS coupling before surgery tended to show a decrease in lesion–GS coupling during recovery (R^2^ = 0.63; *p* = 0.002; Figure 5B). We also observed a significant negative association between Δ$\hat{\beta }$ and the total number of cognitive deficits acquired during recovery (Figure 5C). Thus, individuals showing the greatest decrease in lesion–GS coupling during recovery (negative Δ$\hat{\beta })$ were more likely to have a larger number of newly acquired cognitive deficits following surgery (positive Δ Total cognitive deficits, R^2^ = 0.38, *p* = 0.03). 

### 3.5. Canonical Resting-State Networks Are Preserved in Brain Tumour Patients

Finally, we tested the hypothesis that functional disruption specifically affects canonical resting-state networks. The integrity of the canonical resting-state networks was assessed in brain tumour patients by calculating the average correlation within each of the seven functional networks and comparing them to the equivalent values from HCs. In HCs, canonical networks had a correlation profile ranging from low values within the limbic network (previously described as a network with low SNR and poor reproducibility [43]) to high values within the visual network (Figure 6). Brain tumour patients had a correlation pattern that did not significantly differ from HCs, with all participants exhibiting values lying within the HC range for all seven networks. Within-network correlations in the hemisphere containing the tumour were similar to those observed in the contralateral hemisphere (non-parametric Wilcoxon test; all corrected *p* > 0.95; Figure 6). Surgical resection did not substantially alter this pattern either. Within-network correlation in the affected hemisphere was also not significantly different before and immediately after surgery (non-parametric Wilcoxon test; all corrected *p* > 0.85; Appendix A). Overall, the presence of the tumour and its resection do not have strong effects on BOLD brain dynamics reflected by the absence of alterations to the canonical resting-state networks, suggesting that tumours induce a change to BOLD dynamics, which may be permanent.

## 4. Discussion

A growing literature is revealing that gliomas not only disrupt the tissue immediately surrounding the tumour but also exert long-range influences on distant brain areas [44,45,46]. Understanding how brain tumours integrate within brain circuits is crucial for prognoses. In this study, we combined longitudinal MRI tumour patient data with normative data from unaffected individuals to determine whether patients’ recovery is related to the effect that gliomas and the lesioned brain have on the GS. We found increased tumour incidence in brain regions with lower coupling with the GS. Moreover, the GS was coupled with the tumour BOLD signal, and its topography remained disrupted both before and after surgical resection. This lesion–brain coupling during recovery was associated with cognitive outcomes. Altogether, these results suggest that the effect the tumour and lesioned brain exerts on GS topography influences, or is a marker of, a patient’s cognitive recovery. 

The GS has been traditionally considered a nuisance effect and is often removed during fMRI pre-processing. Nevertheless, spectroscopy [47], electrophysiological [11] and interventional [48] studies have demonstrated that the GS also contains neuronal components. Despite several efforts [49], there is still no consensus regarding whether the algorithmic attenuation of physiological and motion-related noise is worth the removal of these neuronal components [10,50,51]. Replicating the prior literature [8,15], we observed a heterogenous GS topography pattern with higher β in the medial occipital cortices and low β in association cortices in HCs. More interestingly, we found an association between the GS and tumour incidence. Although the origin of glioma is still a matter of debate, it has been hypothesised that oligodendrocyte precursor cells (OPCs) are the cellular source of this type of tumour [52], which is supported by the fact that gliomas can be transformed into cancer cells through experimental manipulation [53]. We have recently shown that glioma incidence is higher in regions populated by OPCs, such as the temporal and frontal cortices [29]. On the contrary, excitatory and inhibitory neurons, which are directly associated with the GS [11], show a different distribution pattern, with decreased populations in medial temporal and frontal cortices [54]. Thus, the negative correlation between tumour incidence and regional coupling with the GS may reflect the differential cell organisation of the underlying tissue. Alternatively, but not mutually exclusively, we have also shown that glioma incidence is higher in regions with high functional connectedness regardless of tumour grade [29]. This preferential tumour localisation follows intrinsic functional connectivity networks, possibly reflecting tumour cell migration along neuronal networks that support glioma cell proliferation [55]. This has been experimentally supported by Venkatesh and colleagues, who showed that stimulated cortical slices promoted the proliferation of paediatric and adult patient-derived glioma cultures [56]. It has been proposed that the hijacking of the cellular mechanisms of normal CNS development and plasticity may underly the synaptic and electrical integration into neural circuits that promote glioma progression. For example, neuron and glia interactions include electrochemical communication through bona fide AMPA receptor-dependent neuro-glioma synapses [57]. These glutamatergic neurogliomal synapses drive brain tumour progression, partially via influencing calcium communication in cell networks connected through tumour microtubules [58]. The coupling between the glioma BOLD signal and the GS described here may be driven by these neurogliomal synapses that integrate cell networks facilitating the synchronisation of tumoural and non-tumoural cells. Nevertheless, we found that glioma activity has less dependency on the GS than the contralateral (healthy) hemisphere. This may be mediated by increased neuronal activity induced by the tumour [59], which, presumably, is abnormally desynchronised from the GS. However, further research will be necessary to explore this hypothesis. 

Psychiatric conditions, such as schizophrenia [60,61] and major depressive disorder [62], induce alterations in GS topography. However, the impact of neurological conditions on the GS is less well known. Here, we describe, for the first time, alterations in GS topography in brain tumour patients that are also preserved after resection and during recovery. Using a similar approach, Li et al. (2021) recently reported an analogous GS topography disruption in patients with idiopathic generalised epilepsy, who presented with reductions in β also in bilateral occipital cortex [63]. Based on evidence from acute stroke patients [64], it has been proposed that deficits in blood perfusion delay the BOLD signal. However, the neuronal, vascular and alternative physiological mechanisms behind this BOLD fMRI and GS disruption are still a matter of debate. 

Focal lesions have been traditionally associated in neurology with ‘focal’ clinical deficits. Nevertheless, brain tumour patients, especially those with gliomas, often present multimodal cognitive deficits that cannot be explained by a focal disruption of their brain function [2,41], which is unsurprising given the infiltrative nature of gliomas [65]. Widespread topological reorganisation of brain functioning has been reported in glioma patients before surgery [66,67]. Here, we found that the coupling between the GS and glioma was significantly higher than with CSF and WM and that it was reduced after surgical resection, suggesting a functional integration of glioma into neural circuits. Accordingly, glioblastoma patients present functionally connected voxels within the tumour mass, although with reduced connectivity strength when compared with HCs [25]. A recent study has found that functionally connected regions within a tumour are enriched for a glioblastoma subpopulation that exhibits a distinct synaptogenic and neurotrophic phenotype [46]. Although BOLD fMRI has been successfully exploited to quantify tumour oxygenation [68], microvascular components [68], tumour delineation [20] and vascular disruption [69], interpreting a BOLD signal from tumour and lesioned tissue can be challenging. Thus, tumours disrupt the complex cellular and chemical neurovascular coupling mechanisms between neuronal firing and cerebrovascular dilatation [70]. As BOLD fMRI is only sensitive to the cerebrovascular response, it is not possible to untangle the contribution of this potential lesion-induced neurovascular uncoupling with this technique. Nevertheless, alterations of brain dynamics in tumour patients have been also observed using electrophysiological imaging techniques, such as MEG [71], which has also shown a high degree of spatial congruence with fMRI for the motor mapping of glioma patients [72]. However, given the major impact that gliomas have on vascular regulation [73], we cannot discard mediation by the blood supply or metabolic alterations in the GS–tumour interactions reported here. 

The use of fMRI for the presurgical mapping of speech, movement and sensation has been associated with improved patient outcomes [16,17,18]. However, the impact of tumour surgery and treatment on cognition has been systematically underestimated [74]. Incorporating objective measurements for monitoring, predicting and ultimately protecting cognition and mental health is a pressing concern for maintaining patients’ quality of life [75]. By comparing tumours with different molecular profiles, it has been hypothesised that slow-growing tumours may allow more time for neuroplastic reorganisation, which increase the recruitment of remote brain areas in the ipsi- and contra-lesional hemispheres [76], not only improving survival rates but also protecting neurocognitive functioning [77]. Here, we observed that the normalised tumour–GS ($\hat{\beta }$) coupling was correlated with the normalised lesion–GS ($\hat{\beta }$) coupling in the post-operative period, which is, in turn, correlated with cognitive recovery. We have also shown preserved functionality in areas around the tumour using electrocorticography during surgical intervention in some of these patients [26]. There is recent evidence in support of the impact that glioma–brain interactions have on cognition. Patients with gliomas functionally connected with the rest of the brain according to MEG had lower scores in auditory and picture naming tasks [46]. Despite this promising evidence, the clinical benefit for patients is still very limited.

We hypothesised that cognitive deterioration may be mediated by alterations in canonical resting-state networks. With this in mind, we have recently shown that the structural integrity of canonical networks is associated with memory recovery [78]. Functional alterations of the default mode network have been associated with poor cognitive performance before glioma surgery [79]. It has been suggested that whereas cortical plasticity is generally high (except around the pre- and post-central gyrus), functional compensation of white matter connectivity is rather low [80]. However, here, we observed that canonical networks have similar within-network correlation values in glioma patients to in healthy controls. Accordingly, a widespread coupling has been reported between glioblastoma functional connectivity and canonical networks, without preferential impact on any of them [25]. The resilience of canonical networks is also supported by studies in hemispherectomy adults who have an equivalent within-network correlation to controls [81]. It has been proposed that neuromodulators may play a prominent role in disrupting the GS without targeting specific network components. For example, the inactivation of the nucleus basalis of Meynert in two monkeys, the principal source of widespread cholinergic and GABAergic projections to the cortex, lead to a strong suppression of GS components, while traditional resting-state networks retained their spatial structure [48]. Nevertheless, further research is needed to better understand the role of individual canonical networks in brain dynamics and cognitive recovery.

The research and clinical communities are exploring how to best incorporate advanced neuroimaging and connectomics into biomarkers of patient prognosis for improved treatment outcomes. A growing number of surgical teams are already including fMRI as a standard procedure for designing better surgical strategies that preserve motor [82] and, in some cases, language [83] functioning. Nevertheless, the distributed nature of high-order cognitive functions across spatially extended brain networks [84] represents an additional challenge for delimiting the areas that can be optimally resected. For that reason, whole-brain imaging approaches are particularly promising for understanding the impact of the tumour on brain functioning. The association between glioma–brain coupling and cognitive recovery reported here could be incorporated into surgical planning as individual risk factors of cognitive deterioration. However, given the non-invasive and complementary nature of techniques aiming to improve surgical outcomes, there is a pressing need for determining how to optimally combine and operationalise these markers. Several strategies have been proposed to translate multi-modal markers into clinical practice, including neuronavigation devices [31], 3D models [85], real-time image-guided surgery [86] and virtual/augmented reality [87]. Current imaging research in neuro-oncology is still based on observational studies with limited sample sizes, while randomised interventional controlled trials for assessing the clinical utility of these approaches are still scarce. A major difficulty of complex interventions such as brain surgery is that traditional double-blind trial strategies for assessing treatment, safety and efficacy are not feasible. For that reason, the IDEAL (Idea, Development, Exploration, Assessment, Long-term study) framework [88] proposes a set of stages for gradually progressing research towards evidence-based treatment that could be the vehicle for translating these imaging-based markers into routine clinical decision making. 

### Limitations

With 55 scans and 31 exhaustive neurocognitive assessments, this study presents the most densely acquired longitudinal dataset of diffuse gliomas and the first analysis on the effect of tumour and lesioned brains on GS and cognition. However, in order to acquire such dense data, the overall sample was reduced to 17 patients, which in turn represents a somewhat limited sample size to understand the heterogeneous effects of conditions such as brain tumours. This reduces the generalisability of the results and increases the chances of reporting non-significant associations. Moreover, although scans were acquired up to four times per patient, only two neuropsychological assessments were administered. Two main constraints impeded the acquisition of cognitive assessment with each MRI scan: (i) potential learning effects when four assessments are conducted serially in a one year period and (ii) the logistical challenge of administering a 2–3 h interview in a hospital setting as part of each patient’s clinical pathway. Beyond demographic and histopathological tumour variability, treatment was decided based on clinical criteria, with 5 patients having only surgical intervention and 12 patients having varied chemo-radiotherapy regimes. All patients had the pre-operative imaging appearances of a diffuse glioma (non-enhancing and without oedema or mass effect); however, subsequent pathological examination revealed a range of histological diagnoses (Appendix A). Although our models regressed out the effect of age, tumour volume and treatment, the limited sample size constrains our ability to discriminate the contribution of these factors to the reported associations. Importantly, some of the key findings establishing associations at the individual level (e.g., tumour–GS coupling and correlation within canonical networks) were replicated for all patients except one.

## 5. Conclusions

Our findings reveal that glioma occurrence is associated with GS topography, which is, in turn, disrupted in brain tumour patients during recovery. Tumour and lesioned brains were coupled with the GS, which was associated with patients’ cognitive recovery, finding no evidence of disruption of canonical resting-state networks. Altogether, these results highlight the potential of exploiting BOLD fMRI to better understand the effect that gliomas and their treatment have on brain dynamics and their potential for patient prognosis.

## Figures and Tables

**Figure 1 cancers-13-05008-f001:**
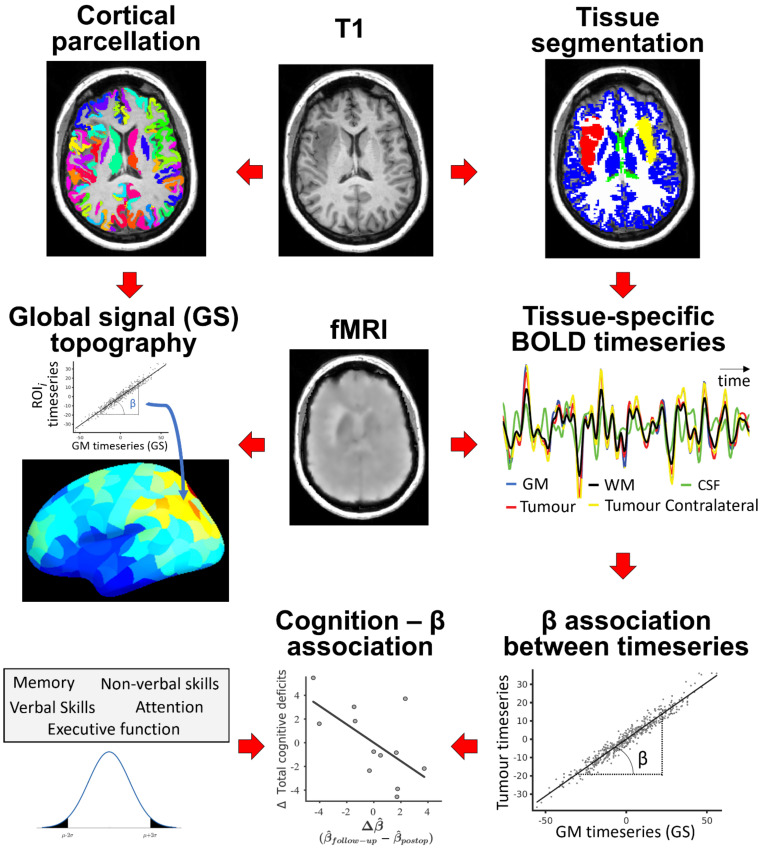
Flowchart of analysis. After segmenting the brain tissues and parcellating the cerebral cortex, BOLD signals were extracted for each tissue type and cortical parcel. The association, β, was computed as the slope relating the time-series of tissue types or parcels with the GS (each dot represents a timepoint). The number of acquired cognitive deficits after surgery (Δ Total cognitive deficits) was correlated with changes in normalised β ($\hat{\beta }$) during recovery, Δ$\hat{\beta }$.

**Figure 2 cancers-13-05008-f002:**
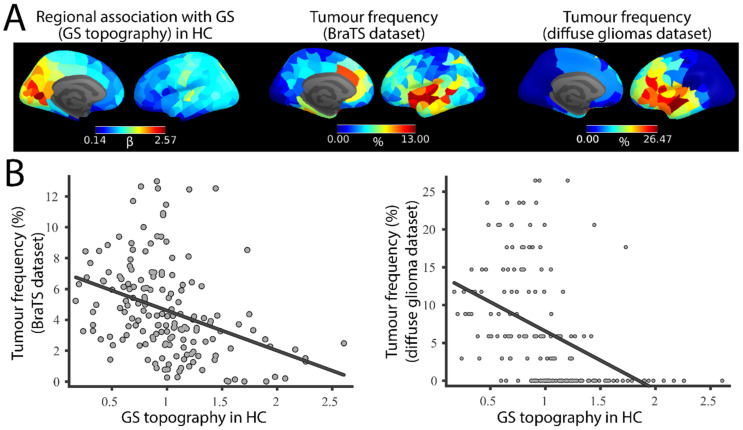
Spatial correlation between GS topography and tumour frequency. (**A**). GS topography is associated with tumour occurrence. (**B**). Associations between GS and regional signals (GS topography) derived from HCs (left) and tumour occurrence in the BRATS (middle) and diffuse glioma (right) datasets.

**Figure 3 cancers-13-05008-f003:**
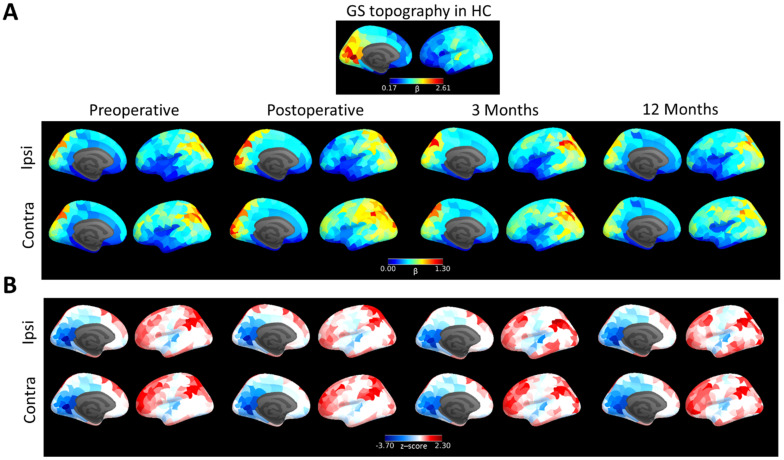
GS topography of brain tumour patients. (**A**). GS topography of HCs compared with brain tumour patients scanned before surgery (leftmost column), within a week after surgery (second column) and 3 months (third column) and 12 months follow-up (rightmost column) for the hemisphere containing the tumour (ipsi) and the contralateral healthy hemisphere (contra). (**B**). GS topography of brain tumour patients normalised to HC values (z-scores; blue represents reduced regional coupling with GS in patients compared with HCs).

**Figure 4 cancers-13-05008-f004:**
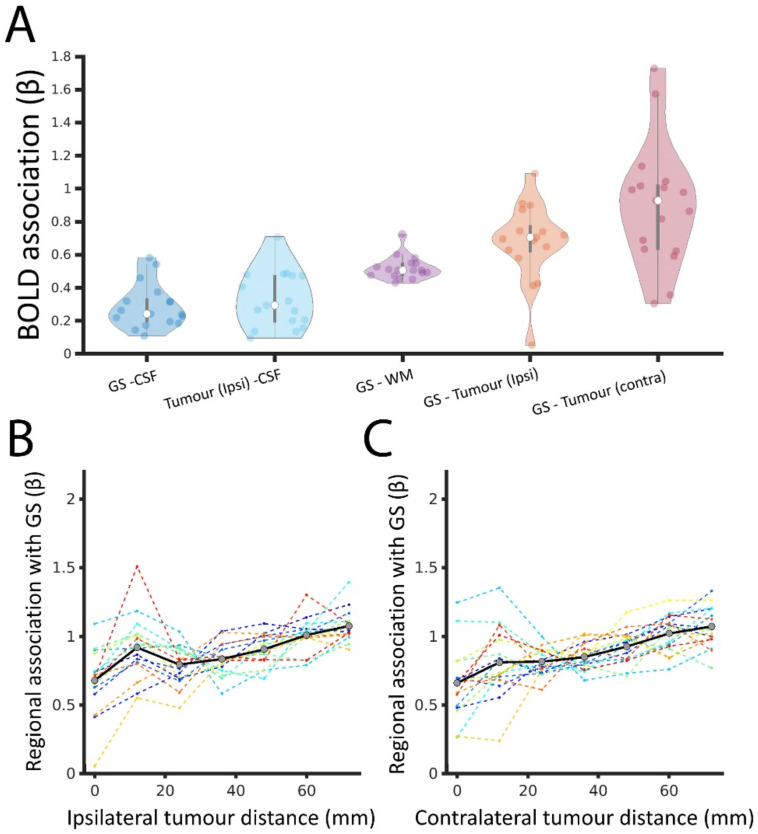
GS coupling with BOLD signal derived from tissue compartments. (**A**). Distribution of β across brain tumour patients (represented by individual dots) between BOLD signals derived from different tissue compartments: grey matter (that is, from which the GS is extracted), cerebrospinal fluid (CSF), white matter (WM), tumour (tumour ipsi) and cortical regions contralateral to the tumour (tumour contra). In the latter case, the β was calculated using a GS estimation that excluded the corresponding voxels contralateral to the tumour (to avoid overlapping between the independent and dependent variables). (**B**). The association, β, between the GS and non-tumour regions as a function of regional tumour distance (mm). (**C**). The association, β, between the GS and non-tumour regions as a function of regional distance (mm) to the contralateral tumour regions (i.e., zero represents homologous regions to the tumour in the contralateral, unaffected, hemisphere).

**Figure 5 cancers-13-05008-f005:**
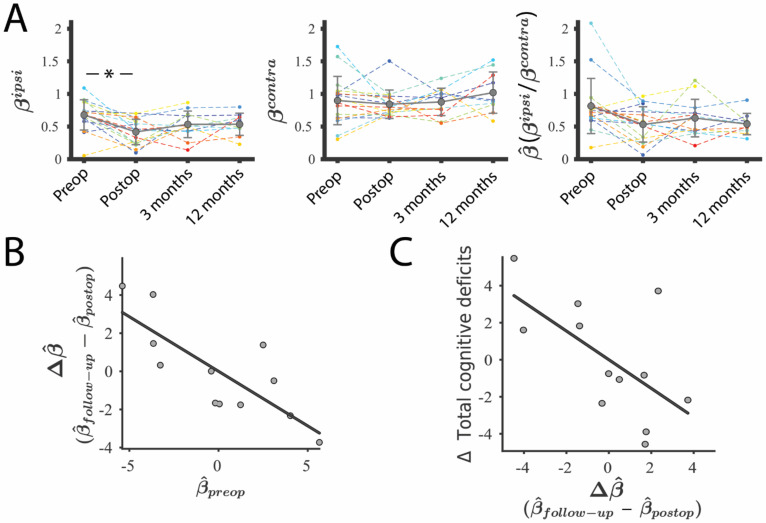
Coupling between GS and lesioned tissue during patients’ recovery. (**A**). Coupling between the GS and tumour (pre-operative) and lesion (post-operative and follow-up) BOLD signal ($\beta^{ipsi}$; left). Coupling between GS and the healthy regions contralateral to the tumour/lesion ($\beta^{contra}$; middle). The normalised coupling was defined as the ratio between both metrics ($\hat{\beta }$; right). Preop, Pre-operative assessment; Postop, post-operative assessment. * represents *p* < 0.05 (**B**). Association between normalised pre-operative tumour–GS coupling (${\hat{\beta }}_{preop}$ ) and the rate of change of the lesion–GS coupling during recovery (Δ$\hat{\beta }$, defined as ${\hat{\beta }}_{follow-up}$ − ${\hat{\beta }}_{postop}$ ). (**C**). Cognitive recovery (positive represents acquired deficits during recovery) as a function of the rate of change of lesion–GS coupling during recovery. Associations were calculated after regressing out the effects of age, tumour volume and type of treatment. ${\hat{\beta }}_{follow-up}$ corresponds with the β value of the last MRI scan available for each patient (i.e., 3 or 12 months, depending on missing data).

**Figure 6 cancers-13-05008-f006:**
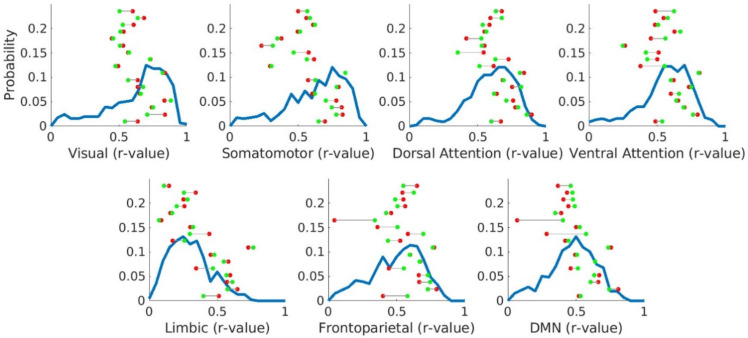
Average correlation within canonical networks in HCs and brain tumour patients before surgery. Blue lines represent the distribution of the average correlation within each of the 7 canonical networks for 653 HCs. Individual points illustrate the within-network correlation for each brain tumour patient when considering the hemisphere containing the tumour (red points) and the hemisphere contralateral to the tumour (green points). The *Y*-axis represents probability values only for HCs, not for the individual points.

## Data Availability

In accordance with ethics requirements, data will be made available to collaborating centres upon request.

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
