# Peer review of "BOLD Coupling between Lesioned and Healthy Brain Is Associated with Glioma Patients’ Recovery"

_cancers, 2021, doi:10.3390/cancers13195008_

Round 1

Reviewer 1 Report

The manuscript submitted explores the functional changes in the brain before and after surgery in glioma patients. It uses a unique datasets made up of both functional and anatomical MRI, as well as instrument data exploring cognition. Overall, the manuscript is well written and contains sound ideas that are important to the CNS cancer community. The reviewer recommends publication with minor revisions. 

Minor Revisions:

1) The layman description is not as easy to follow as the abstract. The authors should consider rewriting the section to be more concise (e.g. number of scans can be removed).

2) In the methods section, it would benefit the manuscript to have a flow chart of the patients used and their exclusion (After exclusions and losses to follow-up, 55 MRIs and 34 neuropsychological assessments from diffuse glioma patients were included in the analyses.). This could be added to the supplemental. 

3) In the methods section, please spell out the abbreviation of your instruments (e.g. AMIPB = Adult Memory and Information Processing Battery Task) when they are described.

4) In the discussion, the authors need to better explore their own findings in the manuscript in the 2nd paragraph. For example, the authors mention the work of Michelle Monje but does not put their results in this manuscript into the context of these findings. The authors found that lower GS was associated with likelihood of tumor? Wasn't GS lower between the tumor and region near when compared to the contralateral side, "the value of β be-tween the GS and tumour signal was significantly lower than that between the GS and BOLD signals derived from the cortical regions contralateral to the tumour (P=0.015; FDR-corrected)"?   

Reviewer 2 Report

This study examined functional imaging in adult glioma patients at various time points before and after surgery, which had the specific aim to test functional interactions between tumour and healthy tissues over time. The primary strength of this manuscript is the robust, longitudinal design, which can answer questions about cause and effect in a vulnerable group of patients. Concerns are related to applicability of these results to clinical settings, particularly because the primary aims of the study are unclear and many analyses are conducted with a relatively small sample size. Comments about various aspects of the manuscript are outlined below.

Introduction

  1. In the introduction, please describe the primary and secondary aims of the study. It appears that the primary aim is to relate global signal to tumour presence, but this is unclear. This also makes it challenging to follow the methods, results, and conclusions.
  2. Pg 3: it is hypothesized that “changes in the topographic relationship of the GS will be related to changes in cognition experienced by patients as a result of their therapy.” However, cognitive testing was only performed in the pre- and post-surgery stage, and not up to 12 months like the MRI scans. Why was longer term cognitive testing not performed for this study?

Methods

  1. Pg 3: criteria of tumour being located “in or near eloquent areas of the brain, i.e., regions thought to be important for speech and executive functions”. Can you describe more details of how this was defined? For example, the cerebellum is also involved with language and cognitive functioning, so were cerebellar tumours also eligible for this study? Similarly, why not include all brain tumours given the focus on global signal? A tumour located in occipital cortex could theoretically impact only visual functioning, but it could impact the networks of the brain and therefore global signal.
  2. Pg 3: the authors report that 17 patients were recruited for the study. Please indicate how many patients were approached vs. consented to the study. Are there any differences in demographic or clinical details in those who consented or not?
  3. Pg 5: the definition of the global signal is challenging to follow. It appears that all tissue types contribute to the global signal as described on Pg. 5: “BOLD signals were extracted and averaged across voxels for several tissue compartments: i) tumour/lesion, as defined by the semiautomatic delineation procedure (referred as ‘Tumour Ipsi’), ii) Cerebrospinal Fluid (CSF), iii) White Matter (WM), iv) brain tissue contralateral to the tumour (referred as ‘Tumour Contra’), and v) Grey Matter (GM) excluding the tumour/lesion, constituting the GS. However, Pg. 6 then indicates that GS was “defined as the average BOLD signal across all GM voxels, excluding tumour”. Is GS all tissue types or just GM (non-tumour) voxels?

Discussion

  1. How can these results be related to interventions? If there are relationships between global signal and tumour presence and/or cognitive functioning, what is the next step for the clinic?
  2. Suggestions for future research are not included in the discussion. Please add these to help readers less familiar with this topic.
